# Gradient-Pattern Micro-Grooved Wicks Fabricated by the Ultraviolet Nanosecond Laser Method and Their Enhanced Capillary Performance

**DOI:** 10.3390/mi15010165

**Published:** 2024-01-22

**Authors:** Guanghan Huang, Jiawei Liao, Chao Fan, Shuang Liu, Wenjie Miao, Yu Zhang, Shiwo Ta, Guannan Yang, Chengqiang Cui

**Affiliations:** 1State Key Laboratory of Precision Electronic Manufacturing Technology and Equipment, Guangdong University of Technology, Guangzhou 510006, China; ghan.huang@outlook.com (G.H.); liaojiawei0404@163.com (J.L.); fanc_001@163.com (C.F.); lshuang000@163.com (S.L.); 18369147071@163.com (W.M.); zhangyu@gdut.edu.cn (Y.Z.); 2State Key Laboratory of Advanced Materials and Electronic Components, Fenghua Advanced Technology Co., Ltd., Zhaoqing 526020, China; tashiwo@china-fenghua.com

**Keywords:** enhanced capillary performance, ultraviolet nanosecond laser, capillary gradient wick, micro-grooved wick

## Abstract

Capillary-gradient wicks can achieve fast or directional liquid transport, but they face fabrication challenges by traditional methods in terms of precise patterns. Laser processing is a potential solution due to its high pattern accuracy, but there are a few studies on laser-processed capillary-gradient wicks. In this paper, capillary step-gradient micro-grooved wicks (CSMWs) were fabricated by an ultraviolet nanosecond pulsed laser, and their capillary performance was studied experimentally. The CSMWs could be divided into three regions with a decreasing capillary radius. The equilibrium rising height of the CSMWs was enhanced by 124% compared to the non-gradient parallel wick. Different from the classical Lucas–Washburn model describing a uniform non-gradient wick, secondary capillary acceleration was observed in the negative gradient direction of the CSMWs. With the increase in laser power and the decrease in scanning speed, the capillary performance was promoted, and the optimal laser processing parameters were 4 W-10 mm/s. The laser-enhanced capillary performance was attributed to the improved hydrophilicity and reduced capillary radius, which resulted from the increased surface roughness, protrusion morphology, and deep-narrow V-shaped grooves induced by the high energy density of the laser. Our study demonstrates that ultraviolet pulsed laser processing is a highly efficient and low-cost method for fabricating high-performance capillary gradient wicks.

## 1. Introduction

In recent years, with the development of aerospace technology [1], high-speed servers [2], and the miniaturization of electronic devices [3], the heat generated within integrated circuits has significantly increased, posing great challenges to thermal management. Heat pipes play an important role in the thermal management of IC chips due to their high reliability and desirable capacity to cool extremely high heat flux [4]. The wick structure is functional in the transport of working fluid, and it can ensure the rapid backflow of working fluid to the evaporator. Hence, the wick structure is an essential part of heat pipes or other two-phase heat transfer devices. Micro-grooves, owing to their high permeability, low flow resistance, and cost-effectiveness, are one of the most commonly used wick structures [5,6]. Currently, the principal approaches to further enhance the capillary wicking capability of micro-grooves mainly involve the development of advanced manufacturing methods to form special surface morphologies [7,8,9] and the optimization of capillary structure design [10].

There are various methods for fabricating micro-grooved wick structures. Traditional methods include chemical etching [11], sintering [12], extrusion [13], micro-machining [14], and sintering-cutting composites [15]. Recently, advanced pulse laser processing technology has shown its advantages of high efficiency, high precision, great processing flexibility, and a wide range of applicable materials (including silicon chips [16], sapphire [17], metals [18], etc.), demonstrating the great potential in the preparation of micro-grooved wick structures. Li et al. [19] studied the fabrication process of micro-grooves on nanocarbon foam using laser etching. It was found that the groove morphology was composed of randomly distributed carbon nanotubes, which enhanced the capillary performance of the nanocarbon foam. Jiang et al. [20] employed a two-step nanosecond laser texturing method to manufacture a chemical-free ultrathin aluminum groove wick, demonstrating the enhancement effect of dual-scale micro-grooves on capillary performance. Xie et al. [21] created a supercapillary surface with uniformly arranged micro-grooves on aluminum by femtosecond laser processing, and the best capillary performance was observed at laser energy densities of 18.49 and 52.67 J/cm^2^. Jiang et al. [22] investigated the influences of spot overlap rate on micro-groove depth uniformity at lower laser energy densities, and it was revealed that a spot overlap rate of 80% could result in higher ablation efficiency and better depth uniformity. Wang et al. [23] used in-situ sintering and laser ablation to fabricate multi-scale groove-like porous silicon carbide wick structures, which presented enhanced capillary performance. Long et al. [24] prepared supernarrow-deep micro-grooves with a groove width of 50 μm and a depth-to-width ratio of 3.5 with high laser scanning speeds, and the groove wicks demonstrated high permeability and transport capacity. Overall, the above studies mainly focus on the influences of laser parameters on the capillary performance of parallel non-gradient micro-grooves but rarely involve capillary-gradient micro-grooved wick structures.

Capillary-gradient wick structures are a type of wick with increasing or decreasing capillary pressure along the axis, which can be induced by surface wettability gradients or structural capillary radius gradients [25]. As presented in Figure 1b, capillary gradients can be classified into two types: continuous gradients (like a slope) and step gradients (like steps). The directions of capillary gradients include negative direction and positive direction, and the negative one is defined as the direction from the side with a larger effective capillary radius to that with a smaller one. Thus, in the negative gradient direction, the contact angle or the capillary radius decreases, resulting in an increase in capillary force; conversely, in the positive gradient direction, the Laplace pressure becomes negative, and liquid transport is unachievable. Therefore, capillary gradient wick structures offer unique advantages in terms of rapid capillary transport [26] and unidirectional fluid transport capability [10]. Particularly, unidirectional transport wicks enable heat pipes to transfer heat in only one direction, exhibiting their potential applications in challenging heat diodes.

The manufacturing methods and performance research of capillary gradient wick structures have attracted extensive attention. For instance, Zhou et al. [27] used sintered copper wire mesh to create a novel 2 mm thick ultra-thin isothermal plate with a radially capillary-gradient wick structure, significantly enhancing the liquid transport capacity within the capillary structure. Zhang et al. [28] employed a diffusion bonding process to fabricate a copper porous micro-mesh wick structure with gradient porosity, which achieved rapid liquid replenishment during boiling and further enhanced the heat transfer coefficient and critical heat flux. Jiang et al. [29] used hot micro-embossing to process a pyramid-shaped vertical capillary gradient structure, enhancing thin-film evaporation and capillary pumping processes simultaneously. Liu et al. [30] used electroplating on an isothermal plate to manufacture wettability-gradient capillary wicks with a gradient capillary pressure direction from the periphery to the center. It was found that the surface wettability of the capillary wicks changed from hydrophilicity to superhydrophilicity, effectively improving the overall performance of temperature uniformity. Xie et al. [31] employed a pulsed fiber laser to process micro-grooves with varying widths and effective capillary radius, and the adjusted radius gradient enhanced the capillary force of the copper substrate. In general, the above methods have poor pattern controllability and low processing efficiency. Fortunately, precise laser processing offers the advantages of high pattern flexibility, high accuracy, good controllability, and low cost when working with non-uniform, gradient, or patterned wick structures. However, there have been a few reports on laser processing to fabricate capillary gradient wick structures, especially groove-type gradient wick structures.

In this study, capillary step-gradient micro-grooved wicks (CSMWs) were fabricated by laser processing. The infrared thermography [32] method was employed to characterize the capillary performance parameter (Δ*P_cap_·K*) of the CSMWs, and the influences of laser parameters (laser power and scanning speed) on the capillary performance were investigated. Our study verifies the unidirectionality of capillary transport in gradient micro-grooved wick structures and also demonstrates the enhancement of capillary performance by laser processing on copper surfaces. Furthermore, this study identifies the optimal laser processing parameters for achieving the best capillary performance and reveals the affecting mechanisms of laser parameters on the capillary performance of the CSMWs in terms of shape, size, morphology, and surface wettability. Finally, the study quantitatively establishes the coupling relationship between laser parameters and capillary performance parameters of CSMWs, providing a valuable basis for selecting laser processing parameters to achieve desirable capillary performance.

## 2. Materials and Methods

### 2.1. Fabrication of Copper CSMWs by Ultraviolet Nanosecond Laser

In this study, an ultraviolet nanosecond laser was used to fabricate micro-grooves with capillary step-gradients on a copper plate with dimensions of 0.8 mm in thickness, 5 mm in width, and 60 mm in length. The copper plate was made of high-purity copper (99.9% Cu) due to its high thermal conductivity and ease of processing. Its thermophysical properties are shown in Table 1. The laser processing principle is illustrated in Figure 1a. The incident laser beam is focused onto the material surface, and the material is heated by the laser beam to reach the ablation threshold. After rapid heating, melting, evaporation, ionization, evaporation spraying, and fluid ejection, the surface material was removed [33,34].

The CSMWs are divided into three regions: the triple-groove region, the double-groove region, and the single-groove region, as shown in Figure 1c. In the negative capillary gradient direction, the total groove width decreases from 60 to 20 μm, with a gradually decreasing capillary radius. Taking the CSMW processed at 2.26 W and 90 mm/s as an example, we can calculate that the capillary diameter (the same as the hydraulic diameter) along the axial direction decreases from 25.12 (triple-groove) to 22.16 μm (double-groove), and then further decreases to 19.20 μm (single-groove). Here, the hydraulic diameter is defined as the ratio of four times of the cross-sectional liquid area in saturated grooved wick to the wetting perimeter of the groove. The decrease in hydraulic diameter facilitates the increase in capillary force in the late stage of capillary rise, promoting secondary capillary acceleration.

The laser spot size and processing positions in different regions of the CSMWs are presented in Figure 1c and Figure 2. The lengths of single-groove, double-groove, and triple-groove regions are 30, 15, and 15 mm, respectively. The specific groove dimensions of the three regions are labeled in Figure 2. As can be seen, the single-groove region has a groove spacing S1 of 40 μm (Figure 2a). For the double-groove region (Figure 2b), the spacing between the neighboring double grooves (noted as S2) is 60 μm, and the distance between the centerlines of the adjacent grooves (noted as a) is 20 μm. In the triple-groove region (Figure 2c), the spacing between neighboring sets of triple grooves (noted as S3) is 80 μm. The total grooved widths of the single-groove region, double-groove region, and triple-groove region are 20 μm, 40 μm, and 60 μm, respectively.

Table 2 lists the basic parameters of the laser system. A 355 nm ultraviolet nanosecond pulse laser source (FOTIA-355-3-30-W, Inno Laser, Shenzhen, China) was employed, and the scanning speed was controlled using a galvanometric scanner (RAYLASE MS-II-10, Raylase AG, RAYLASE GmbH, Weßling, Germany). The energy distribution of the laser spot followed a Gaussian distribution, resulting in V-shaped grooves. To investigate the influence of laser parameters, a total of 12 samples were prepared by laser processing. The corresponding processing parameters are listed in Table 3. Among these samples, nine were subjected to a capillary rise test in the negative gradient direction, and three were subjected to a capillary rise test in the positive gradient direction (marked as “PG”). The copper plate surfaces for laser processing were fully polished, followed by ultrasonic cleaning with deionized water. Subsequently, an acid treatment process was employed to remove oil stains and insoluble substances from the copper plate surfaces, followed by cleaning and drying using deionized water.

### 2.2. Surface Characterization

The surface morphology of the CSMWs was characterized using a scanning electron microscope (SEM, HITACHI SU8220, HITACHI, Tokyo, Japan) at an emission voltage of 15 kV. The element distribution of the laser-induced micro-grooves was measured using an energy-dispersive X-ray spectrometer (EDS, Bruker XFlash 6|30, Bruker, Karlsruhe, Germany). Contact angle measurements were conducted on double-groove regions. The contact angles were measured by a contact angle measurement instrument (CSCDIC-200S, SINCIN, Dongguan, China) with an accuracy of ±1°. A water droplet of 2 μL was used for contact angle measurements, and the contact angles at five different points on each sample were measured to obtain the average value.

### 2.3. Capillary Rise Test by IR Camera

The capillary rise test of the CSMWs was conducted in both negative and positive gradient directions, and the schematic diagram of the test apparatus is shown in Figure 3. In the test, the infrared thermography method [32] was used to characterize the capillary rise rate of the CSMWs. Ethanol with high wetting capacity was applied as the working fluid, and its thermophysical properties are shown in Table 1. Three samples processed with the same laser power were fixed to a metal bracket for simultaneous testing. The experimental setup was enclosed in a box with black body walls to minimize interference from ethanol evaporation and external temperature fluctuations. A Fotric227s-L23 infrared camera with a resolution of 512 × 384 was employed, and the meniscus could be clearly captured by the infrared camera due to the emissivity difference between the working fluid and the copper material. The camera was operated at a frame rate of 24 frames per second, providing a time accuracy of 41.6 ms. The test time for each set of samples was 3 min, and the ambient temperature was maintained at 26 ± 0.2 °C throughout the test. The distance between the samples and the camera was 200 mm, with a rise height uncertainty of less than 0.3 mm. The capillary rising height is determined by the average height of the meniscus, which is indicated by the boundary of purple and orange colors. The operating criteria are presented in Appendix A.

## 3. Capillary Theory and Data Reduction in Capillary Performance Parameter

The capillary pressure induced by the curved liquid–gas interface can be expressed by the Laplace–Young equation [36]:(1)$${\Delta P}_{\mathrm{cap}}=\frac{2\sigma \cos\theta }{r_{p}}$$
where *r_p_*, *σ*, and *θ* are the capillary radius of the pore, surface tension, and contact angle, respectively. When the effective capillary radius (*r_eff_*) is used to represent the ratio of *r_p_* and *cosθ*, Equation (1) can be rewritten as:(2)$$\Delta P_{\mathrm{cap}}=\frac{2\sigma }{r_{\mathrm{eff}}}$$

When a wick comes into contact with the working fluid, the liquid will spontaneously rise along the wick. Since the wick is placed vertically during the test, there is a balance between gravity and capillary pressure. Then, capillary pressure, friction pressure, and gravity can be related as follows [32,37,38]:(3)$$\Delta P_{\mathrm{cap}}=\frac{\mu \varepsilon }{K}\;h\frac{\mathrm{dh}}{\mathrm{dt}}+\rho \mathrm{gh}$$
where the first term on the right is the viscous friction according to Darcy’s Law, and *ρgh* is the gravitational hydrostatic pressure. In detail, *dh*/*dt* is the rate of capillary rise, *h* is the rising height of the working fluid, *μ* is the viscosity of the working fluid, *ρ* is the density of the working fluid, *g* is the acceleration of gravity, *ε* is the porosity of the wick, and *K* is the permeability of the capillary.

In general, the liquid transport capacity can be evaluated by the capillary performance parameter (Δ*P_cap_·K*) [32,39], which can be obtained from Equation (3):(4)$$\Delta P_{\mathrm{cap}}\cdot K=h(\mu \varepsilon \frac{\mathrm{dh}}{\mathrm{dt}}+\rho \mathrm{gK})$$

The capillary performance parameter (Δ*P_cap_·K*) can be calculated by the following method proposed by Deng et al. [32,39]. In Equation (4), the rising velocity, *dh*/*dt*, can be obtained from the derivative of the capillary rising height versus time. Rearranging Equation (4) yields:(5)$$(\Delta P_{\mathrm{cap}}\cdot K)\frac{1}{h}\;-\;\rho \mathrm{gK}=\mu \varepsilon \frac{\mathrm{dh}}{\mathrm{dt}}$$

When we set *x* = 1/*h* and *y* = *dh*/*dt*, Equation (5) can be rewritten as:(6)$$y=\frac{{\Delta P}_{\mathrm{cap}}\cdot K}{\mu \varepsilon }x-\frac{\rho \mathrm{gK}}{\mu \varepsilon }$$

Through linear fitting by several data points of *x* and *y* from the test, the Δ*P_cap_·K* can be calculated from the slope of the fitting line. The capillary performance parameters of the three regions of the CSMWs are different, so the slopes for different segments of the fitting line are also different. For easy comparison, in this study, the comprehensive capillary performance parameter of the three segments is adopted based on linear fitting for all data.

## 4. Uncertainties

The uncertainty of the copper substrate dimensions is estimated to be 1.5%. The absolute uncertainty of the contact angle measurements is ±1°. The uncertainty of the micro-groove dimensions from laser processing is ±5 μm. The uncertainty of the capillary rise height measured by the infrared camera method is within 1.5%. The uncertainty of dh/dt is estimated to be within 3%. The standard deviations of the capillary performance parameters from linear fitting are presented in Table 4, with a relative standard deviation ranging from 1.51% to 2.21%.

## 5. Results and Discussion

### 5.1. Influences of Laser Parameters on Micro-Groove Morphology

Figure 4 illustrates the cross-sectional morphology of the double-groove regions processed with different laser parameters. It is evident that laser power and scanning speed have significant influences on the cross-sectional morphology of the micro-grooves. As the laser power increases or the scanning speed decreases, the depth of the micro-grooves gradually increases. For example, at a scanning speed of 10 mm/s, when the power increases from 2.26 to 4 W, the depth of the V-shaped groove increases from 43.4 to 50.1 μm; at a certain laser power of 2.26 W, when the scanning speed decreases from 90 to 10 mm/s, the groove depth increases from 30.2 to 43.4 μm. At higher laser power and lower scanning speeds, more material on the copper substrate reaches the ablation threshold, resulting in deeper grooves due to the increased melting and evaporation [40].

Laser-processed micro-grooves exhibit unique cross-sectional morphology features distinct from those fabricated by traditional methods. First, laser processing produces a recast layer and raises the top of the micro-grooves above the original copper surface due to the deposition of molten and evaporated copper material on the top of the micro-grooves. The recast layer increases the depth of the grooves, leading to faster capillary transport. Moreover, the spikes and voids within the recast layer also facilitate extensive evaporation of the working fluid on the meniscus surface [24]. Second, laser-processed grooves have a unique V-shaped geometry with a high aspect ratio owing to the Gaussian distribution of laser spot energy. The high aspect ratio is attributed to the extremely high energy density of the laser (25.46 J/cm^2^). As a result, the depth-to-width ratio of the grooves fabricated in this study (~3.4 in Figure 4a) is significantly greater than that of the grooves produced through traditional extrusion molding (~1.28) [41] or high-speed spinning methods (~1.25) [42]. From another point of view, the laser-caused material addition method (such as laser-induced electrodeposition) [43] is an effective way to enhance surface capillary performance. For example, the laser-induced maskless electrodeposition method can fabricate complex patterns with high flexibility. Thus, it can be used to fabricate precise capillary-enhanced structures. Besides, the morphology of the fabricated surface is rich (such as stray deposition shape and pyramidal structure), which facilitates surface hydrophilicity.

Figure 5 presents the top view of the SEM images of double-groove regions with various laser parameters. As the laser power increases, the surface roughness of the grooves also shows an increasing trend. For instance, when the laser power increases from 2.26 to 4 W at a fixed scanning speed of 90 mm/s, the surface morphology translates from a smooth cement-like texture to a bark-like texture and then to a rock-like texture, with more and more pronounced surface protrusions. On the other hand, when the laser power is fixed, the smoothness of the groove surfaces gradually diminishes with the increase in scanning speed. This phenomenon is particularly pronounced in the sample group fabricated at a laser power of 2 W. As can be seen, the density of protrusions on the recast layer and groove sidewalls of the 2 W-10 V sample is significantly higher than that of the 2 W-90 V sample. This can be attributed to the increased absorption energy density of the material at higher laser power and lower scanning speed, which intensifies heat transfer and diffusion behaviors, leading to more prominent secondary deposition of laser-ablated copper material [34,40]. As a result, the overlapping phenomenon in the recast layer intensifies and the surface roughness increases [44,45].

Since the oxidation of the copper surface affects the contact angle, EDS analysis of the CSMWs was conducted to determine the O element mass fraction. Figure 6 shows the EDS images of single grooves, double grooves, and triple grooves with a laser power of 4 W at a scanning speed of 10 mm/s. All samples were exposed to the air for approximately 1 h before testing. The EDS images reveal a pronounced distribution of oxygen element in the recast layer, indicating the oxidation of copper during laser processing. Notably, due to altitude differences, the oxygen element in the central region of the V-shaped groove is not captured, presenting a black color. Figure 7 illustrates the mass fraction change trends of copper and oxygen elements as a function of total groove width. The mass fraction of the oxygen element is strongly related to the total groove width, and the oxidation effect becomes more pronounced as the total groove width increases. This is because the increase in total groove width results in a larger total copper surface exposure to air, leading to a more intense degree of oxidation [20].

### 5.2. Influences of Laser Parameters on Contact Angle

Figure 8a–i shows the top-view SEM images of double-groove regions fabricated with different laser powers at different scanning speeds. In the images, the axis of the groove is along the left-right direction. It is obvious that the hydrophilicity of laser-induced surfaces is significantly enhanced in comparison to that of the unprocessed surface. As shown in Figure 8j, the original contact angle of the unprocessed surface is 93.1 ± 1°, which is a little hydrophobic due to mild oxidation after short-term storage before testing. The three CSMW samples fabricated with a laser power of 4.00 W present the best hydrophilicity, with the contact angle ranging from 8.9 ± 1° to 17.3 ± 1°. Those fabricated with a laser power of 3.12 W present worse hydrophilicity (with the contact angle ranging from 10.3 ± 1° to 33.3 ± 1°), while the samples fabricated with a laser power of 2.26 W show the worst hydrophilicity (with the contact angle ranging from 41.2 ± 1° to 58.0 ± 1°). As shown in Figure 8a, the contact angle achieves the minimum value of 8.9 ± 1° when the sample is fabricated with a laser power of 4 W at a scanning speed of 10 mm/s. Hence, it can be concluded that the wettability is improved with increasing laser power and decreasing scanning speed, which facilitates the enhancement of capillary performance [23].

### 5.3. Enhanced Capillary Performance of the CSMW Compared to the Non-Gradient Parallel Wick

Figure 9 presents the enhanced capillary performance of the CSMW compared to a non-gradient parallel micro-grooved wick. The capillary rising rate and height of the gradient wick substantially outperform the parallel one. As shown in Figure 9a, the rising curve of the parallel wick became stable at a height of 24.0 mm, while the one of CSMW presented two times of secondary accelerations and finally reached about 53.8 mm. The equilibrium rising height of the CSMW was enhanced by 124% compared to the parallel one. The capillary performance parameter was promoted from 9.061 × 10^−9^ N to 9.81 × 10^−9^ N, with an improvement percentage of 8.2%. The enhancing mechanism is promoted, balancing between capillary force and permeability in various flowing stages. The CSMW provide a large permeability at the early stage and a large capillary force at the final stage, which promotes capillary flow efficiency.

### 5.4. Influences of Capillary-Gradient Direction on Capillary Performance of the CSMWs

To verify the unidirectionality of capillary transport in the CSMWs, the influences of capillary-gradient direction on the capillary rise performance were investigated. Figure 10 presents the SEM images of the CSMWs. As can be seen, the total groove width as well as the capillary radius decrease along the negative gradient direction. Additionally, the grooves are continuous in the vertical direction since the longitudinal grooves are processed in the final step, preventing any recast layer blockage in the longitudinal direction. Figure 11 depicts the capillary rise height of the sample group fabricated with a laser power of 4 W as a function of time in both negative and positive gradient directions. Figure 12 displays the IR images of the equilibrium capillary rise height of the above samples after capillary rise for 3 min.

The CSMWs experimentally exhibit unidirectional liquid transport capacity. In the CSMW with a positive gradient, the working fluid meniscus fails to cross the boundary line (at a height of 30 mm, as shown in Figure 12b) between the single-groove and double-groove regions. This is because liquid transport will be blocked by negative Laplace pressure induced by increasing the capillary radius in the positive gradient direction, which is known as the capillary blocking phenomenon. On the other hand, in the negative gradient direction, the working fluid can cross the boundary lines of neighboring regions, moving from the triple-groove region to the single-groove region, as shown in Figure 12a. These results indicate that the gradient direction significantly affects capillary rise performance. The equilibrium capillary rise heights for the 4 W-10 V, 4 W-50 V, and 4 W-90 V samples under negative gradient conditions are 83%, 55%, and 38%, which are higher than those under positive gradient conditions, respectively, with an average increased percentage of 58.6%.

The CSMWs with a negative gradient exhibit a unique secondary capillary acceleration phenomenon, which contributes to enhanced capillary performance. Taking the 4 W-10 V example as an example, its capillary rise rate does not conform to the classical Lucas–Washburn model [46] for a uniform capillary wick. On its capillary rise curve (Figure 11), there is a secondary acceleration segment while crossing the region boundary, even at the end of the wick. This is due to the sudden reduction in the capillary radius of the CSMW, which produces an increased capillary force and additional driving force. As a result, the equilibrium rising height of the 4 W-10 V sample can reach 55 mm in the negative gradient direction, which is significantly greater than the height (30 mm) in the positive gradient direction for the same sample.

### 5.5. Influences of Laser Parameters on Capillary Performance of the CSMWs

#### 5.5.1. Influence of Laser Parameters on Capillary Rise Height

Figure 13 and Figure 14 illustrate the capillary rise process and the capillary equilibrium rise heights of the CSMWs fabricated with nine combinations of laser parameters, respectively. The meniscus in all samples increases rapidly in the early stages. The secondary acceleration phenomenon was observed at both the 15 mm height interfaces of the 3 W samples group and the 15 mm and 30 mm height interface of the 4 W samples group. Such a capillary rise height curve of the CSMWs is unique and different from the classical Lucas–Washburn modeling curve for uniform wick with a continuously reducing slope [47,48].

The capillary rise height significantly increases with the increase in laser power. Given a fixed scanning speed of 10 mm/s, when the laser power increases from 2.26 to 3.12 W and then to 4 W, the equilibrium capillary rise height increases from 15.6 to 27.5 mm and then to 55 mm, as shown in Figure 14. As the laser power increases, deeper and narrower V-shaped grooves can be produced, facilitating higher capillary pressure. Additionally, the rough and folded surface morphology resulting from the increased laser power can effectively reduce the contact angle and thus increase the Laplace pressure.

When the laser power is fixed, the capillary rise height significantly increases with the decrease in scanning speed. For any given laser power, the samples fabricated at a scanning speed of 10 mm/s exhibit a higher capillary rise height than those fabricated at the other two higher speeds. Taking the sample group fabricated with a laser power of 4 W as an example, the equilibrium height for the scanning speed of 10 mm/s (55 mm) is higher than that for 50 mm/s (42 mm) or 90 mm/s (34 mm). Furthermore, with the increase in laser power, the sensitivity of capillary rise height to scanning speed becomes more pronounced. For instance, the height difference is relatively small for the samples fabricated with a laser power of 2.26 W at various scanning speeds, but it is significantly larger for those fabricated with a laser power of 4 W. The laser scanning speed affects the laser pulse density on the material surface. When the laser scanning speed is lower, the amount of raw material removed is increased, resulting in a larger structural depth [24]. For ideal V-shaped micro-grooves, the capillary rise rate increases with the increase in structural depth.

#### 5.5.2. Coupled Effect of Laser Parameters on Capillary Performance Parameters of the CSMWs

Figure 15 illustrates the coupled effect of laser power and scanning speed on the capillary performance parameter (Δ*P_cap_·K*). With the increase in laser power and the decrease in scanning speed, the capillary performance parameter shows an increasing trend. The 4 W-10 V sample exhibits the highest capillary performance parameter of 9.81 × 10^−9^ N, which is 293% higher than that of the 2 W-90 V sample (2.49 × 10^−9^ N). The mechanisms of the enhanced capillary performance of the 4 W-10 V sample can be attributed to the deepest groove depth (50.1 μm), the smallest contact angle (8.9 ± 1°), the roughest surface structure, and the most abundant surface protrusions (as shown in Figure 5). For a new pattern, the optimal parameters in this study are applicable because this parameter produces the best surface hydrophilicity and a smaller capillary radius, which are generally needed by various patterns.

## 6. Conclusions

In summary, capillary step-gradient micro-grooved wicks on copper substrate were fabricated using an ultraviolet nanosecond pulse laser, and IR thermography was employed to characterize the capillary rise performance of the CSMWs fabricated with different laser parameters. Some conclusions can be drawn, as follows:(1)The increase in laser power and the decrease in scanning speed can deepen V-shaped grooves and increase the surface roughness of CSMWs, resulting in more intricate surface morphology and a smaller contact angle. Besides, the equilibrium capillary rising height of the CSMW was greatly enhanced by 124% compared to the non-gradient grooved wick.(2)The capillary gradient direction of the CSMWs has substantial influences on the capillary performance. In the positive gradient direction, the meniscus experiences capillary blocking at the boundary of neighboring regions. In contrast, for the negative gradient direction, there is a secondary acceleration phenomenon at the region boundary. This can be attributed to the sudden increase in capillary force induced by the decrease in capillary radius, resulting in a unique capillary rise curve that is significantly different from the classical Lucas–Washburn model for uniform capillary wicks.(3)The laser parameters significantly affect the capillary performance of the CSMWs. The equilibrium capillary rise height increases with the increase in laser power and the decrease in scanning speed. The CSMW sample fabricated with a laser power of 4 W at a scanning speed of 10 mm/s has the highest capillary performance parameter of 9.18 ± 0.15 × 10^−9^ N, which is 293% higher than that of the 2 W-90 V sample. Therefore, it is very important to select appropriate laser parameters for the enhancement of the capillary performance of the CSMWs.(4)The enhancement mechanisms of lase processing for the capillary performance of the CSMWs involve improved hydrophilicity and reduced capillary radius. The laser-processed surfaces exhibit increased roughness and abundant protrusion morphology with a recast layer, which can increase the surface area and thereby lower the contact angle, resulting in improved hydrophilicity. The high-energy density and Gaussian distribution of laser energy produce V-shaped grooves, which have elongated the regions at the groove ends, resulting in a large depth-to-width ratio and a reduced capillary radius. To sum up, the increased hydrophilicity and decreased capillary radius result in increased Laplace pressure, thereby enhancing the capillary performance of the CSMWs.

The laser processing technology in this study can provide an efficient and cost-effective method for fabricating patterned wicks with improved capillary performance.

## Figures and Tables

**Figure 1 micromachines-15-00165-f001:**
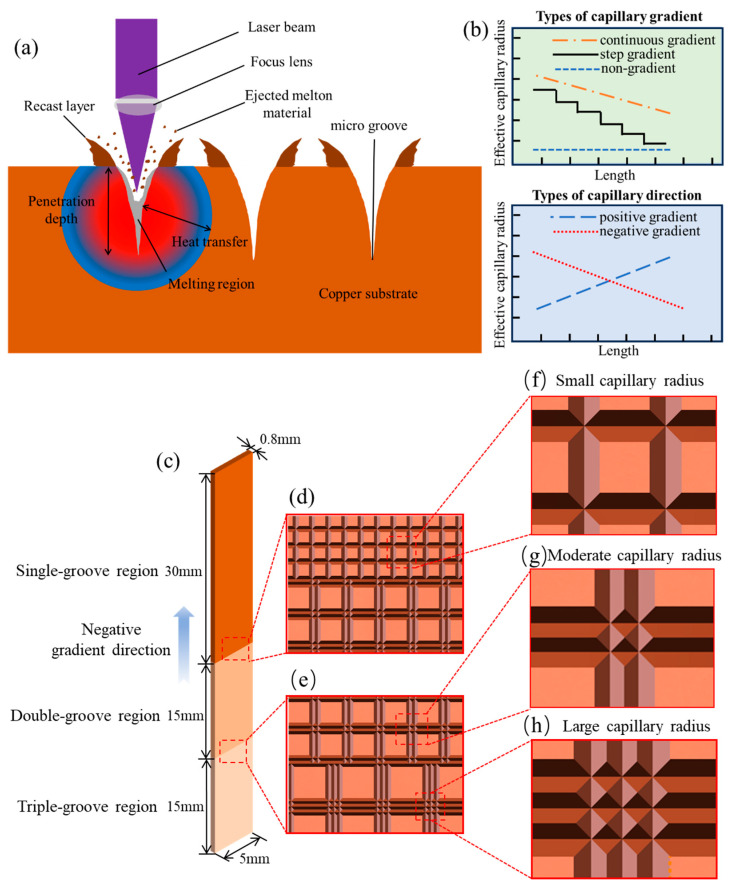
Schematic of CSMWs and their fabrication mechanism by ultraviolet nanosecond laser. (**a**) Fabrication mechanism of high-aspect-ratio V-shaped copper grooves by laser processing. (**b**) Two types of capillary gradient (continuous gradient and step gradient) and two types of gradient direction (positive gradient direction and negative gradient direction). (**c**) Three regions of CSMWs with different capillary radii, including a single-groove region, a double-groove region, and a triple-groove region. (**d**) Junction between the single-groove and double-groove regions. (**e**) Junction between the double-groove and triple-groove regions. Partially enlarged details of (**f**) single-groove region with small capillary radius, (**g**) double-groove region with medium capillary radius, and (**h**) triple-groove region with large capillary radius.

**Figure 2 micromachines-15-00165-f002:**
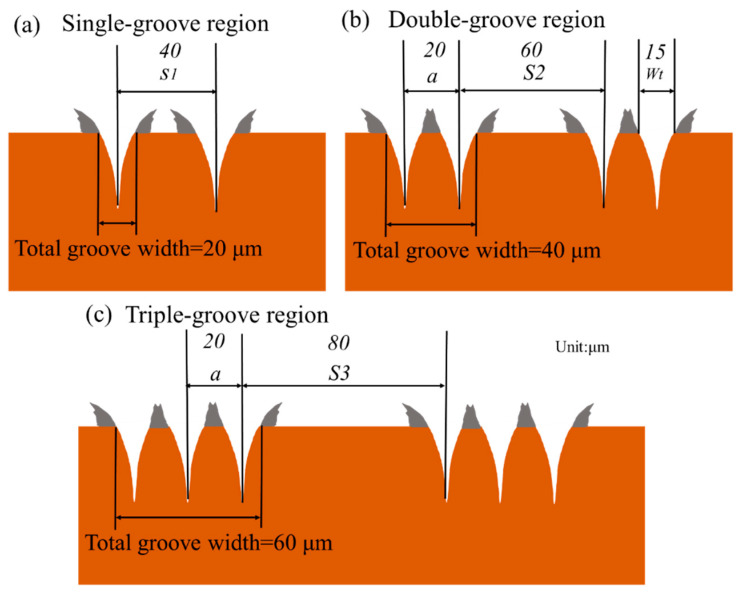
Dimensions of micro-grooves in (**a**) single-groove region, (**b**) double-groove region, and (**c**) triple-groove region.

**Figure 3 micromachines-15-00165-f003:**
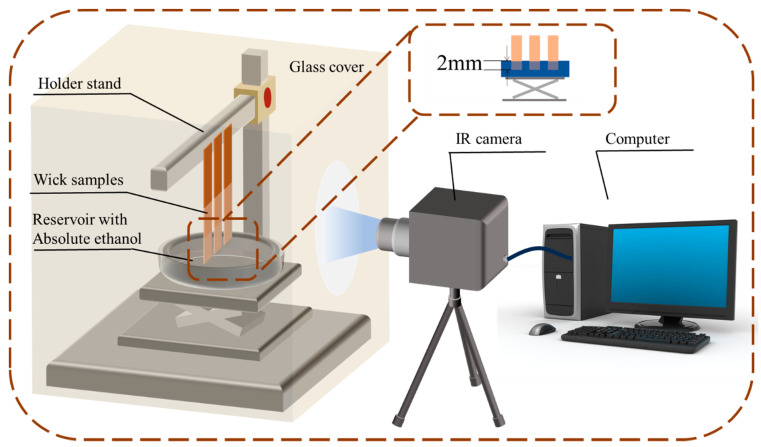
Schematic diagram of the capillary rise test apparatus based on an infrared camera.

**Figure 4 micromachines-15-00165-f004:**
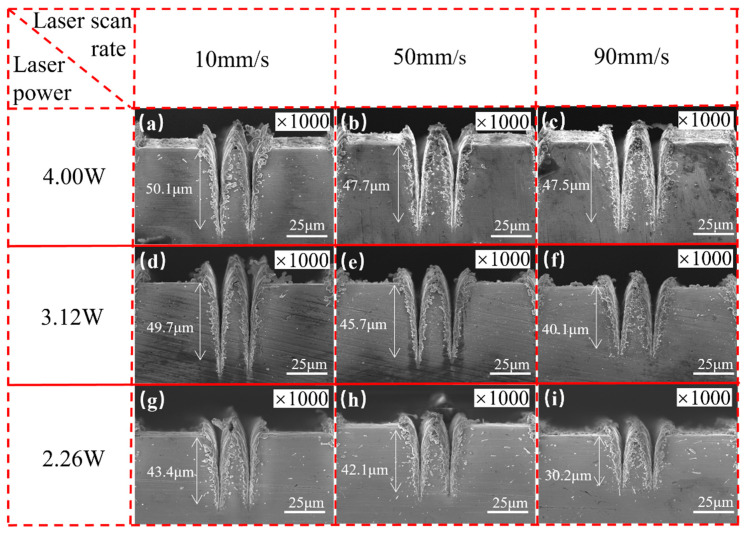
Cross-sectional morphology of double-groove regions fabricated by various laser powers and scanning speeds: (**a**) 4.00 W-10 mm/s, (**b**) 4.00 W-50 mm/s, (**c**) 4.00 W-90 mm/s, (**d**) 3.12 W-10 mm/s, (**e**) 3.12 W-50 mm/s, (**f**) 3.12 W-90 mm/s, (**g**) 2.26 W-10 mm/s, (**h**) 2.26 W-50 mm/s, and (**i**) 2.26 W-90 mm/s.

**Figure 5 micromachines-15-00165-f005:**
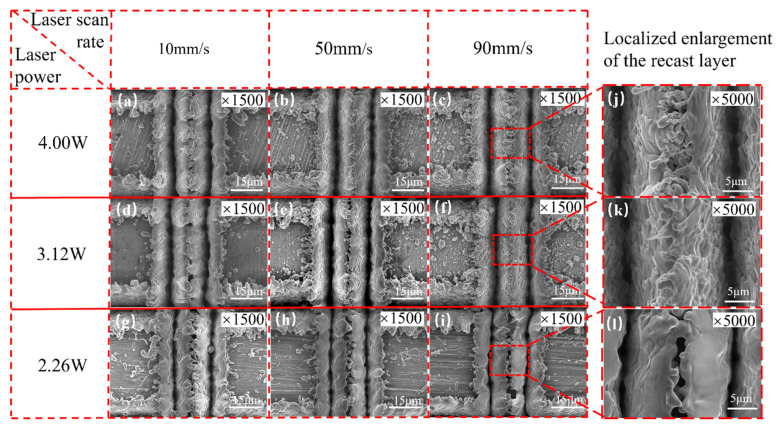
Top-view SEM images of double-groove regions fabricated with different laser powers at different scanning speeds: (**a**) 4.00 W-10 mm/s, (**b**) 4.00 W-50 mm/s, (**c**) 4.00 W-90 mm/s, (**d**) 3.12 W-10 mm/s, (**e**) 3.12 W-50 mm/s, (**f**) 3.12 W-90 mm/s, (**g**) 2.26 W-10 mm/s, (**h**) 2.26 W-50 mm/s, and (**i**) 2.26 W-90 mm/s. Partially enlarged views of the recast layer with a laser power of (**j**) 4.00 W, (**k**) 3.12 W, and (**l**) 2.26 W at a scanning speed of 90 mm/s.

**Figure 6 micromachines-15-00165-f006:**
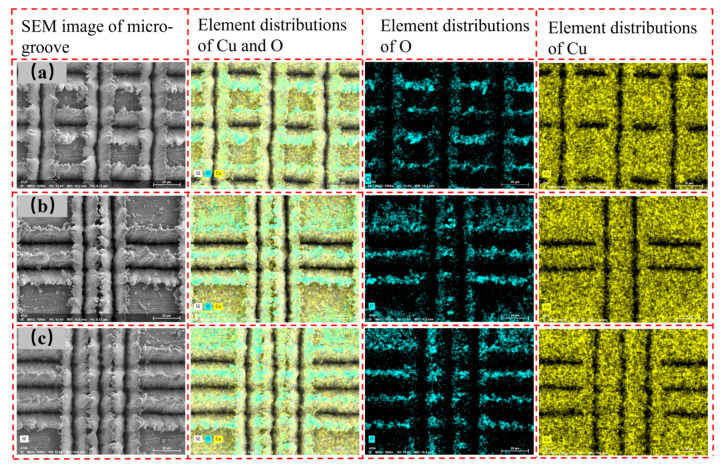
EDS element distribution in micro-groove regions with different total groove widths and capillary radius: (**a**) single-groove regions with a total groove width of 20 μm, (**b**) double-groove regions with a total groove width of 40 μm, and (**c**) triple-groove regions with a total groove width of 60 μm.

**Figure 7 micromachines-15-00165-f007:**
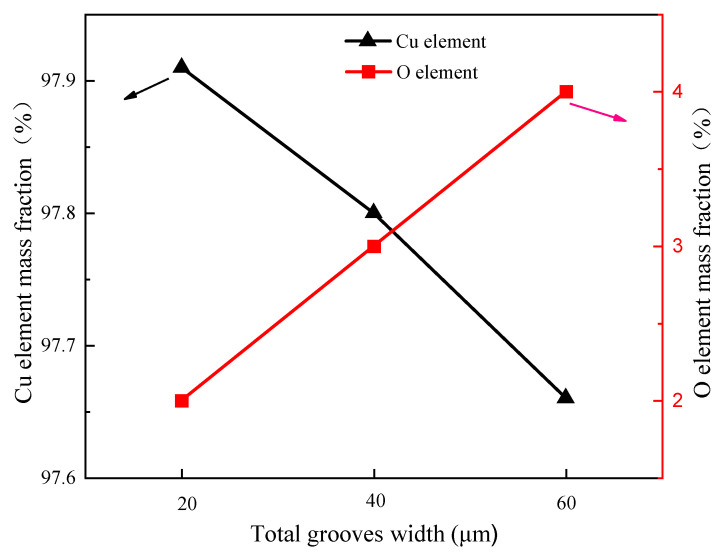
Mass fraction distribution of copper and oxygen elements as a function of total groove width for different micro-groove regions.

**Figure 8 micromachines-15-00165-f008:**
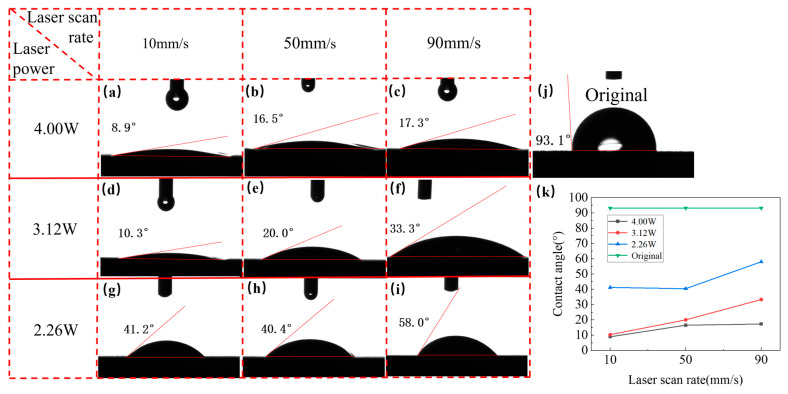
Top-view SEM images of double-groove regions fabricated with different laser powers at different scanning speeds: (**a**) 4.00 W-10 mm/s, (**b**) 4.00 W-50 mm/s, (**c**) 4.00 W-90 mm/s, (**d**) 3.12 W-10 mm/s, (**e**) 3.12 W-50 mm/s, (**f**) 3.12 W-90 mm/s, (**g**) 2.26 W-10 mm/s, (**h**) 2.26 W-50 mm/s, and (**i**) 2.26 W-90 mm/s. (**j**) Original contact angle of an unprocessed copper plate. (**k**) Change the trend of contact angle as a function of scanning speed.

**Figure 9 micromachines-15-00165-f009:**
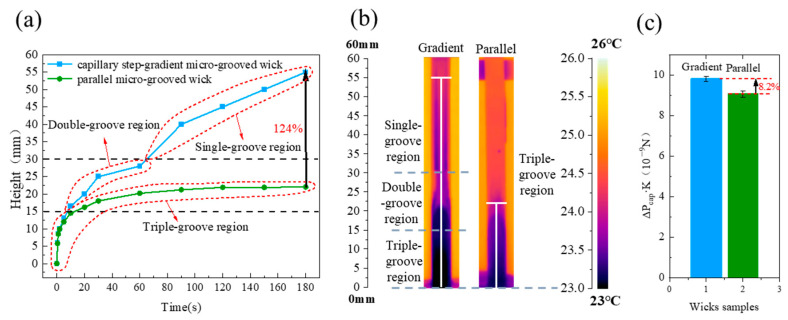
The comparisons between capillary step-gradient micro-grooved wick and parallel micro-grooved wick in terms of (**a**) dynamic capillary rising height, (**b**) equilibrium capillary rising height (in 3 min), and (**c**) capillary performance parameters (Δ*P_cap_·K*). The laser parameters are 4.00 W-10 mm/s, and the parallel wick is only composed of a triple-groove region.

**Figure 10 micromachines-15-00165-f010:**
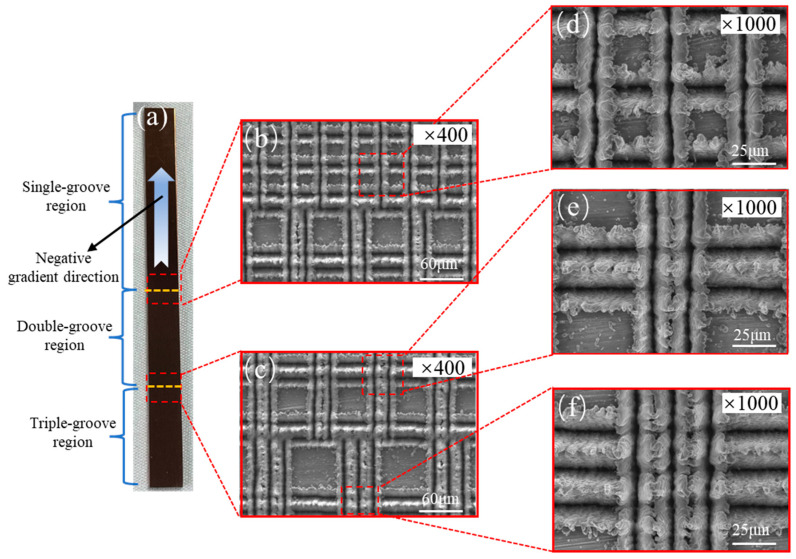
SEM images of the overall CSMW fabricated using an ultraviolet nanosecond laser: (**a**) real image of the CSMW with three regions; SEM images of junctions in (**b**) a single-double groove region and (**c**) a double-triple groove region; and SEM images of (**d**) a single-groove region, (**e**) a double-groove region, and (**f**) a triple-groove region.

**Figure 11 micromachines-15-00165-f011:**
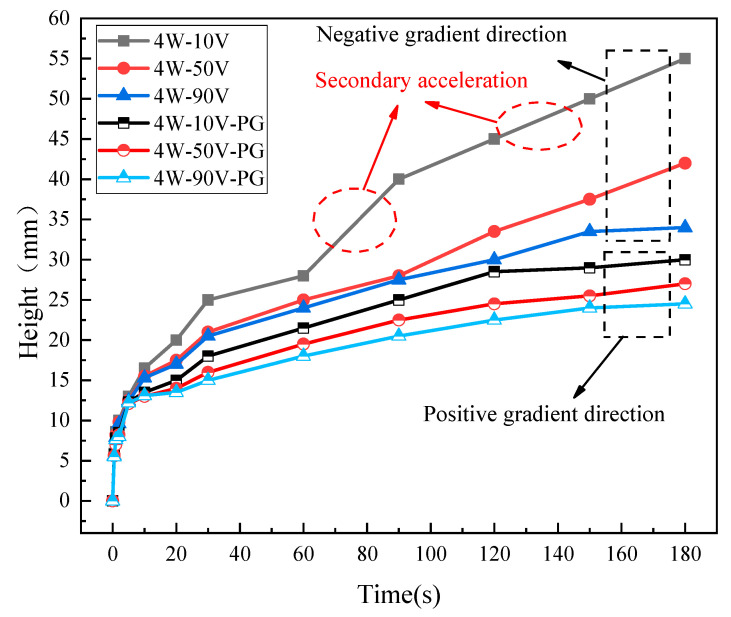
Capillary rise heights of the CSMWs as a function of time in positive and negative gradient directions. The negative gradient direction is from the triple-groove region to the single-groove region with a reduced capillary radius, while the positive gradient direction is the opposite.

**Figure 12 micromachines-15-00165-f012:**
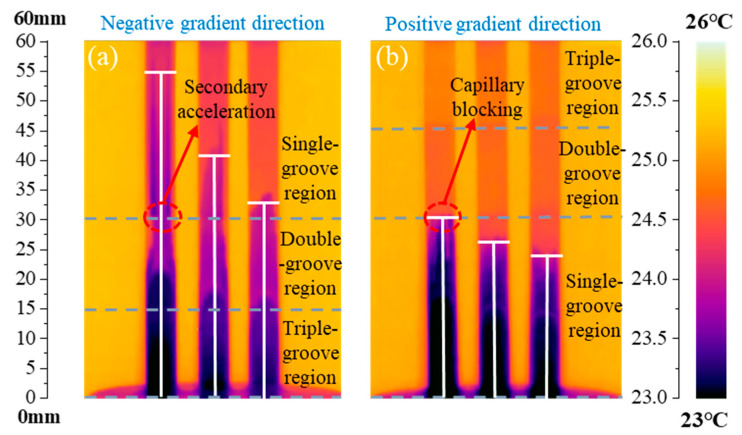
IR images of the equilibrium capillary rise height of the CSMWs in (**a**) negative and (**b**) positive gradient directions. The laser power is 4.00 W, and the scanning speeds are 10, 50, and 90 mm/s from left to right for each subfigure, respectively.

**Figure 13 micromachines-15-00165-f013:**
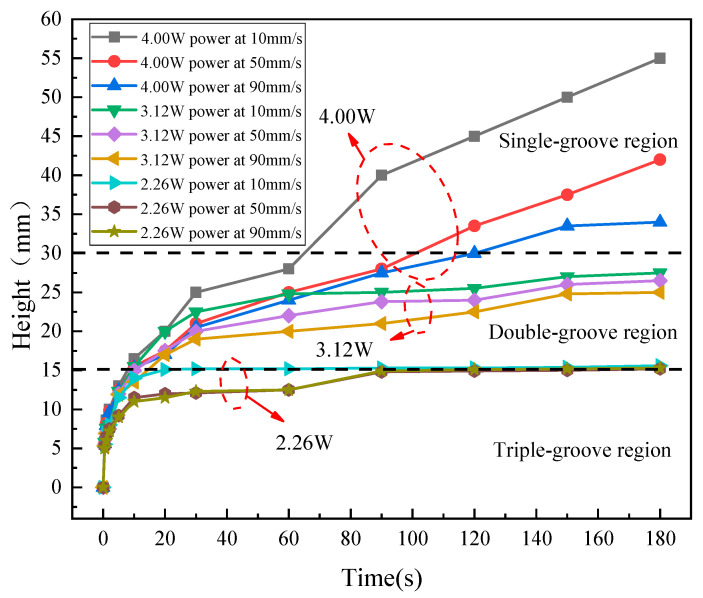
Capillary rise heights of the CSMWs fabricated with different laser powers at different scanning speeds.

**Figure 14 micromachines-15-00165-f014:**
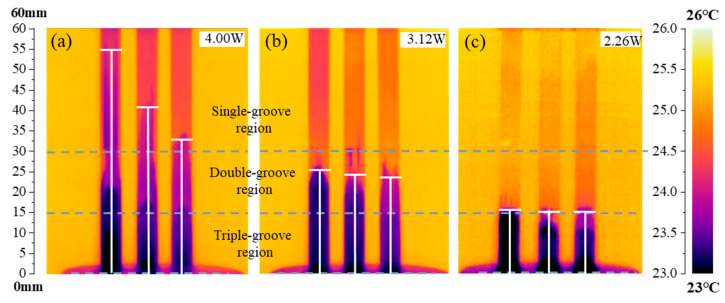
Equilibrium capillary rise heights of the CSMWs fabricated with different laser powers at different scanning speeds: (**a**) 4.00 W, (**b**) 3.12 W, and (**c**) 2.26 W. The laser scanning speeds in each subfigure are 10, 50, and 90 mm/s from left to right, respectively, and the test time is 3 min.

**Figure 15 micromachines-15-00165-f015:**
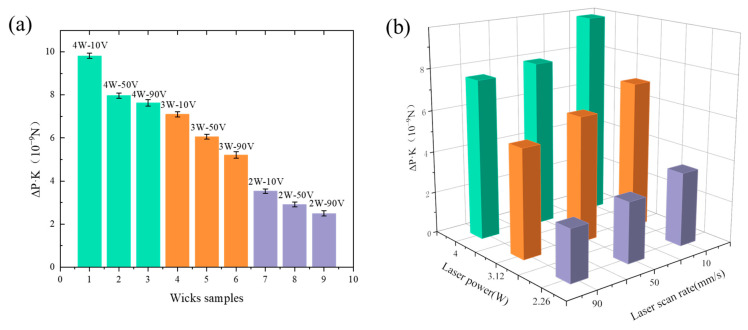
Capillary performance parameters (Δ*P_cap_·K*) of the CSMWs fabricated with different laser parameters in the form of (**a**) Two-dimensional bar chart and (**b**) Three-dimensional bar chart.

**Table 1 micromachines-15-00165-t001:** Thermophysical properties of copper [23] and ethanol [35].

Thermophysical Properties	Copper	Ethanol
Density (g/cm^3^)	8.93	0.79
Surface tension coefficient (N/m)	-	0.022
Viscosity (10^−3^ Pa·s)	-	1.10
Thermal conductivity (W/(m·K))	386	0.1800
Boiling point(K)	-	351.15
Heat capacity (J/(g·K))	0.386	2.44
Thermal expansion coefficient (10^−6^/K)	17.5	250
Thermal diffusivity (mm^2^/s)	111	8.71679 × 10^−8^

**Table 2 micromachines-15-00165-t002:** Characteristic parameters of an ultraviolet nanosecond pulsed laser system.

Parameters	Value	Units
Wavelength	335	nm
Averaged laser power	2.26~4	W
Pulse width	12	ns
Laser frequency	50	kHz
Spot diameter	20	μm
Single pulse energy	80	μJ

**Table 3 micromachines-15-00165-t003:** Laser parameters of the CSMWs.

Wick Sample Code	Laser Power, *P* (W)	Laser Scan Rate, *V* (mm/s)	Scanning Times, *n*
4 W-90 V	4.00	90	5
4 W-50 V	4.00	50	5
4 W-10 V	4.00	10	5
4 W-90 V-PG	4.00	90	5
4 W-50 V-PG	4.00	50	5
4 W-10 V-PG	4.00	10	5
3 W-90 V	3.12	90	5
3 W-50 V	3.12	50	5
3 W-10 V	3.12	10	5
2 W-90 V	2.26	90	5
2 W-50 V	2.26	50	5
2 W-10 V	2.26	10	5

**Table 4 micromachines-15-00165-t004:** Capillary performance parameter (Δ*P_cap_·K*) and its deviation from the CSMW samples.

Wick Samples	Capillary Performance Parameter, Δ*P_cap_·K* (10^−9^)	Standard Deviation of Δ*P_cap_·K* (10^−10^)	Relative Standard Deviation of Δ*P_cap_·K*
4 W-90 V	7.63	1.21	1.59%
4 W-50 V	7.97	1.23	1.54%
4 W-10 V	9.81	1.48	1.51%
4 W-90 V-PG	6.31	1.13	1.79%
4 W-50 V-PG	6.56	1.15	1.75%
4 W-10 V-PG	8.32	1.46	1.75%
3 W-90 V	5.21	0.97	1.86%
3 W-50 V	6.06	1.10	1.81%
3 W-10 V	7.10	1.24	1.74%
2 W-90 V	2.49	0.53	2.12%
2 W-50 V	2.91	0.62	2.13%

## Data Availability

Upon reasonable request, Guannan Yang, the corresponding author, is willing to provide the data that support the conclusions in this paper.

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
