# Peer review of "Gradient-Pattern Micro-Grooved Wicks Fabricated by the Ultraviolet Nanosecond Laser Method and Their Enhanced Capillary Performance"

_micromachines, 2024, doi:10.3390/mi15010165_

Round 1

Reviewer 1 Report

Comments and Suggestions for Authors

This paper has fabricated capillary step-gradient micro-grooved wicks (CSMWs) based on the ultraviolet nanosecond pulsed laser and experimentally studied the capillary performance. The effects of laser parameters on the capillary performance were explored and the enhancement mechanisms of lase processing were demonstrated. This paper is interesting and well written. I recommend publication of this paper after minor revision.

Here are some specific comments:

1)      The physical parameters of the selected solid material and working fluid should be added in a table.

2)      The permeability K is missed in the Eq.(3).

3)      Please add more details about the calculation method of the derived parameters.

4)      What are the optimized scanning speed and laser power to achieve the best capillary performance of a new patterned wick?

Reviewer 2 Report

Comments and Suggestions for Authors

In this manuscript, UV laser is employed to fabricate capillary step-gradient micro-grooved wicks on copper substrate, and the equilibrium capillary rising height of the CSMW has been enhanced by 124% compared to the non-gradient grooved wick, which fully demonstrate the effectiveness the method. The manuscript is well structured, and the experiment is well conducted with amply discussion of the result. Overall, I think the paper can be accepted for publication, while there is one suggestion for consideration or discussion.

1.      Laser is efficient for V-groove etching, as used in this study. Meanwhile, it can be used to induce or enhance deposition of metal A on another substrate B (i.e., https://doi.org/10.1016/j.optlastec.2023.110315). Since different material may be associated with different capillary performance, is it possible to further improve the indicator (such as rising height) via laser-caused material addition?

Reviewer 3 Report

Comments and Suggestions for Authors

In this manuscript, the authors created gradient-pattern micro-grooved wicks on copper plates with nanosecond pulsed lasers. The capillary performance of the obtained samples was characterized by IR thermography and the underlying mechanisms were discussed. The results are interesting to the broad readership of Micromachines. Before this manuscript becomes acceptable, the authors are required to address the following comments well.

1. Please double-check the manuscript and make sure to add SPACE between value and unit.

2. Please add an explanation for l in Eq. (5).

3. In Figure 8(k), why does the original contact angle value become smaller than 93 degrees shown in Fig. 8(j)?

4. For the IR thermography images, what are the criteria for determining the liquid level? In Fig. 12(a), the height of the liquid seems higher than the marked values.

Based on the abovementioned comments, the manuscript is recommended for major revision. A revised manuscript is required.

Round 2

Reviewer 3 Report

Comments and Suggestions for Authors

The authors have answered the comments well. The manuscript is recommended for publication.